# iMedSTAM: Interactive Segmentation and Tracking Anything in 3D Medical Images and Videos

Tobias Friedetzki ⬤, Lorenz Haberzettl, Ricarda Buttmann, Frank Puppe, and Adrian Krenzer ⬤

Julius-Maximilians University of Würzburg, Sanderring 2, 97070 Würzburg, Germany
`tobias.friedetzki@uni-wuerzburg.de`

**Abstract.** The increasing volume of complex 3D biomedical imaging data highlights the need for accurate and efficient analysis methods. Segmentation of such data is essential for diagnosis, anatomical analysis, disease monitoring, and treatment planning. However, existing segmentation algorithms often struggle with the variability of object structures and the diversity of imaging modalities. To address these challenges, we introduce iMedSTAM, a promptable foundation model for 3D image and video segmentation. The model is also capable of progressively improving segmentation quality based on user interactions. iMedSTAM was developed by fine-tuning EfficientTAM on a large-scale dataset comprising over 270,000 3D image–mask pairs and 4,000 video–mask pairs, covering five different medical imaging modalities. In addition, we extend the EfficientTAM architecture with a bidirectional inference and memory mechanism that enables the processing of volumetric data. iMedSTAM significantly outperforms all previous models on the publicly available validation set in the coreset track and achieves state-of-the-art results in the all-data track. On the test set, our model reaches an average final DSC and NSD of 0.805 and 0.842, respectively. For DSC_AUC and NSD_AUC, which measure the cumulative improvement through additional user interactions, iMedSTAM achieves scores of 3.129 and 3.258.

**Keywords:** Interactive Segmentation · 3D Medical Image · SAM2

## 1 Introduction

The segmentation of medical images is an essential component in clinical practice. It supports accurate diagnosis [25], facilitates the analysis of anatomical structures [37], enables disease monitoring [28], and assists in treatment planning [22]. By precisely delineating organs, lesions, and other anatomical structures in medical imaging, clinicians can gain critical insights to provide targeted patient care. Deep learning approaches have revolutionized the field by achieving not only human-level accuracy but also a high degree of automation [1,32,38].

To date, most models in this domain have been developed with a focus on specific organs or lesions, which makes them less effective at handling previously

unseen types of structures and domains. Similar to the developments in natural language processing, where advanced network architectures and large-scale datasets have led to the rise of large language models [30], a shift is also occurring in medical image segmentation. There is a growing trend away from specialized task-specific models toward general-purpose models. Notable examples include the Segment Anything Model (SAM) [12] and its adaptation for medical imaging, MedSAM [17]. These models are capable of segmenting arbitrary objects in 2D images using user-provided spatial prompts that guide the model in identifying the target object.

Although MedSAM has demonstrated strong performance in segmenting various medical structures and across different imaging modalities, it is not well-suited for processing 3D images such as computed tomography (CT) scans, as SAM was originally designed for 2D images. To address this limitation, Gong et al. [6] extended SAM with a 3D adapter, and SAM-Med3D [31] introduced a 3D image encoder. Other interactive segmentation approaches that aim to capture the three-dimensional spatial relationships in volumetric medical images include VISTA3D [7] and SegVol [3]. VISTA3D is trained on CT images and processes them as voxel cubic patches using a sliding window inference strategy. In contrast, SegVol employs a zoom-out–zoom-in mechanism to simplify user interaction with large 3D images.

For interactive segmentation of natural videos, SAM2 [23] has emerged as a state-of-the-art method, and its more efficient variant, EfficientTAM [34], has also demonstrated strong performance. It has been demonstrated that SAM2-like architectures have the potential for application in the medical domain, as shown by Shen et al. [26], Zhu et al. [39], and more recently in MedSAM2 [19].

Although several approaches have already been explored, there remains a lack of general models capable of segmenting both 3D medical images and videos within an interactive process. Most existing models have been trained on only one or a few imaging modalities. Furthermore, the types of prompts supported and the ability to refine segmentations interactively are limited. For example, VISTA3D can only process point prompts, while SegVol exclusively supports box inputs. As a result, iterative refinement through additional user prompts is not possible with SegVol. Only the recently proposed nnInteractive [4], which builds on the nnU-Net framework [10], offers six distinct interaction channels and natively supports a refinement approach.

The "Foundation Models for Interactive 3D Biomedical Image Segmentation" challenge aims to develop universal 3D biomedical image segmentation models that can not only adapt to a wide range of anatomical structures and imaging conditions but also iteratively improve segmentation quality based on user interactions. The challenge provides a large-scale training dataset comprising over 270,000 3D image–mask pairs and 4,000 video–mask pairs, covering five different medical imaging modalities.

To address this challenge, we propose an architecture based on EfficientTAM, as it has already demonstrated strong results on natural videos while offering efficient inference times. Furthermore, since it employs the same prompt encoder

as SAM2, it is capable of processing both box and point prompts. To enable processing of 3D images, we extend the inference pipeline and adapt memory handling across individual slices accordingly. Similar to the approaches of Shen et al. [27] and MedSAM2 [19], we propagate spatial prompts in both directions starting from the central slice. Finally, we fine-tune the model on the large biomedical development set provided by the challenge. Our main contributions can be summarized as follows:

1. We introduce iMedSTAM, a foundation model for interactive segmentation of both 3D medical images and videos, supporting both box and point prompts, as well as iterative refinement.
2. We extend the EfficientTAM architecture with a bidirectional inference and memory mechanism that enables the processing of volumetric data, and fine-tune it on a large-scale medical dataset.
3. We implement an efficient post-processing step that resolves potential overlaps between predicted masks of different classes based on the model's confidence in each prediction.

## 2   Method

### 2.1   Interactive 3D and Video Segmentation Pipeline

For interactive segmentation of volumetric data, we adopt a video-based approach similar to MedSAM2 [19] and Shen et al. [27]. We first slice the 3D image along its third dimension, creating 2D slices that we treat analogously to video frames. This allows us to handle even large 3D input images. Fig. 1 provides an overview of the interactive segmentation pipeline. The model starts from an initial user prompt, which can be either a bounding box or a positive point click at the center of the target object. This prompt always refers to the 2D space of the middle slice. Starting from this middle slice, the model sequentially predicts segmentation masks in both directions. To preserve spatial consistency across slices, the model integrates a memory module that incorporates information from previous slices into the current prediction.

Finally, the individual 2D masks are aggregated into a 3D segmentation mask and presented to the user. The user can review the result and refine it by adding additional points to any slice — positive points in under-segmented regions and negative points in over-segmented areas. With this additional input, the model repeats the segmentation process, enabling iterative refinement. Videos are treated the same as 3D images and follow the same pipeline.

### 2.2   Model

iMedSTAM (Fig. 2) extracts features from individual slices or frames using an image encoder. Instead of passing these features directly to the mask decoder, the model first conditions them on the user input prompts and the embeddings from previous slices. The mask decoder then generates the corresponding 2D segmentation masks.

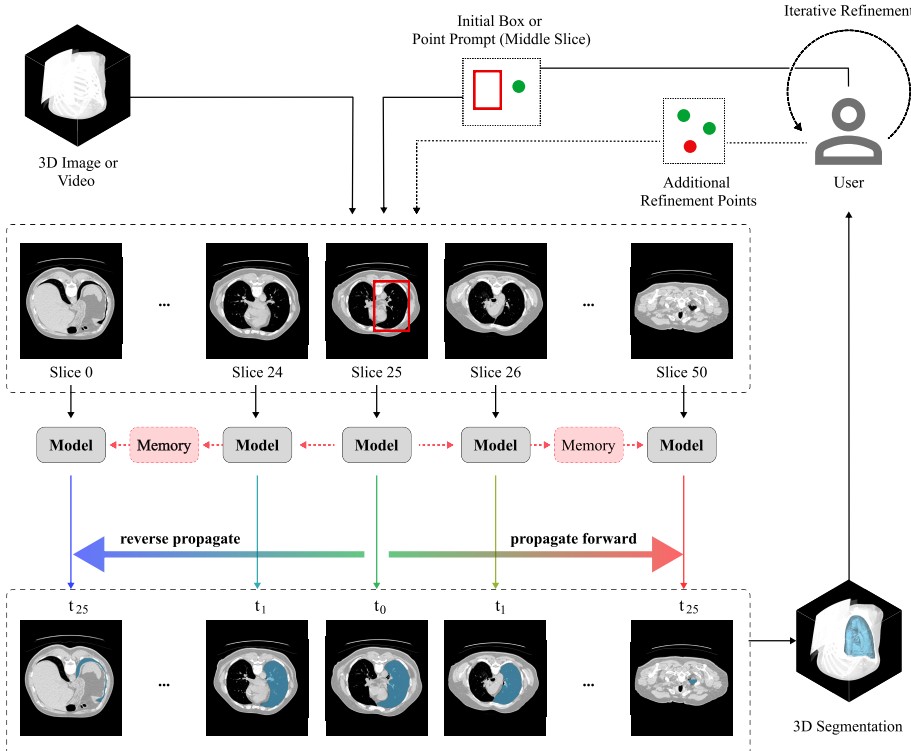

**Fig. 1.** Interactive segmentation pipeline with iMedSTAM. The 3D medical image or video is divided into individual slices or frames. The user initially marks the desired object in the middle slice using a bounding box or a positive point click. Based on this initial prompt, the model sequentially predicts the masks for all slices in both directions from the middle. In subsequent iterative steps, the user can refine the segmentation by adding additional positive or negative refinement points.

The model is based on the EfficientTAM architecture [34], which itself is a more efficient variant of SAM2 [23]. The improved efficiency results from replacing the Hiera [24] image encoder in SAM2 with a lightweight, non-hierarchical ViT [29], and from using a more efficient memory module that approximates spatial memory tokens. Additionally, EfficientTAM includes a pretrained version that operates at an input resolution of $512 \times 512$ instead of $1024 \times 1024$, further reducing computational cost. In our experiments, we observed that EfficientTAM, when used without transfer learning, results in a performance drop of approximately 9% in 3D medical image segmentation compared to SAM2. However, it achieves a twofold reduction in runtime per case (see Table 6 in the appendix). Given that interactive segmentation requires online inference with reasonably fast user feedback, this trade-off is acceptable.

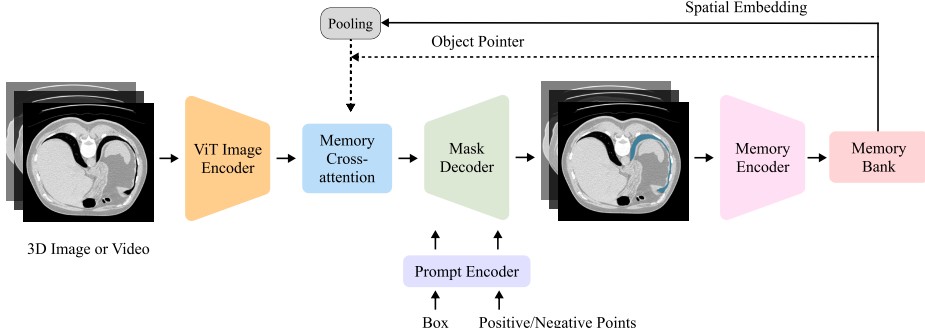

**Fig. 2.** iMedSTAM architecture. The model processes slices or frames sequentially. A lightweight ViT image encoder first extracts features from the current slice. These image embeddings are then corss-attended to memory representations from previous slices. The mask decoder incorporates user input prompts and predicts the segmentation mask for the current slice. Finally, a memory encoder transforms the prediction and image embeddings and stores them in a memory bank for subsequent slices.

**Image Encoder** For the lightweight ViT image encoder, we use a patch size of $16 \times 16$ and a non-overlapping $14 \times 14$ windowed attention mechanism. To efficiently extract features, we employ 6 global attention blocks. The output of the image encoder is a $16\times$ downsampled embedding of the input image. Given an input resolution of $512 \times 512$, the image embedding is therefore $32 \times 32$.

**Memory Module** Since the image encoder processes slices independently, a memory module is required to re-establish a connection along the third dimension (spatial or temporal), enabling consistent object tracking. In the first pass, which is always the middle slice, the memory encoder downsamples the output mask and sums it element-wise with the image features from the image encoder to create spatial embeddings. These spatial embeddings are stored in a memory bank, implemented as a FIFO list that retains information from the last seven slices. In addition to spatial embeddings, a set of lightweight object pointer vectors is stored to represent the current object location.

In subsequent passes, the model conditions the image embeddings of the current slice via cross-attention with both the spatial embeddings and object pointers from previous slices. To reduce computational cost, the model approximates the spatial embeddings with average pooling before applying cross-attention.

Unlike the default implementation of EfficientTAM, we start in the middle and perform both a forward and a backward pass. During memory cross-attention, we therefore only consider entries in the memory bank originating from the current direction. For example, when processing from the middle toward the bottom, the model ignores spatial embeddings and object pointers from the earlier pass from middle to top. This design prevents confusion and ensures that the model maintains strong attention to the initial prompt.

**Prompt Encoder and Interaction Simulation** iMedSTAM supports both box- and point-based prompts. The prompt encoder maps each point to a 256-dimensional embedding by summing its positional encoding with a learned embedding that specifies the prompt type (positive or negative). A box is represented by two points, corresponding to its top-left and bottom-right corners. During training, user prompts are simulated. For the initial prompt, we first identify the middle slice of the target object and generate a bounding box based on the ground truth mask. To improve model robustness, we apply random shifts to the box prompt. In cases where a bounding box is not suitable due to the structure of the object, such as with vessels or myocardium, we use a positive click at the center of the object as the initial prompt instead.

To simulate additional refinement points, we locate the center of the largest error region from the previous prediction. If this region corresponds to an under-segmentation, we place a positive click. Otherwise, we place a negative click. This refinement process is repeated three times per sample during training. Combined with the initial forward pass, this results in a batch size of four per training iteration.

**Mask Decoder** Within the mask decoder, the memory-conditioned image embeddings and prompt embeddings are mapped to a segmentation mask. The mask decoder consists of stacked two-way transformer blocks, followed by an upsampling layer and an MLP that maps the tokens to a linear classifier. This classifier outputs the probability of the object being present at each location. Unlike EfficientTAM, we always condition on all prompts, including refinement points that may refer to future slices or originate from earlier ones. Similar to the memory attention mechanism, we include only prompts associated with slices belonging to the current pass direction.

### 2.3   Coreset selection strategy

To select a reduced development set, we replicate the distribution of the different 3D modalities present in the validation set. Since the training dataset originates from several distinct subsets, we randomly sample from each according to their respective proportions. We ensure that at least one sample is picked from each subset. This strategy guarantees that the model is still exposed to a broad variety of inputs during training.

### 2.4   Post-processing

In a post-processing step, we resize the model output back to the original dimensions of the input image. In cases where multiple objects within a 3D image need to be segmented, we additionally apply a non-overlapping mechanism. This ensures that, in the event of potential mask overlaps, the selected label is not determined by processing order, but rather by the model's confidence in the

presence of each object. This is possible because the model outputs mask logits rather than directly producing binarized masks.

To avoid memory issues, especially in large 3D images or long videos with many objects, we retain only the mask logits of slices still needed for comparison in the non-overlapping mechanism. This is achieved by processing objects in ascending order based on the starting slice of their bounding box. Furthermore, we cache the model state between refinement iterations, which helps to speed up the overall inference process.

## 3   Experiments

### 3.1   Dataset and evaluation metrics

The development set is an extension of the CVPR 2024 MedSAM on Laptop Challenge [18], including more 3D cases from public datasets[1] and covering commonly used 3D modalities, such as Computed Tomography (CT), Magnetic Resonance Imaging (MRI), Positron Emission Tomography (PET), Ultrasound, and Microscopy images. The hidden testing set is created by a community effort where all the cases are unpublished. The annotations are either provided by the data contributors or annotated by the challenge organizer with 3D Slicer [11] and MedSAM2 [19]. In addition to using all training cases, the challenge contains a coreset track, where participants can select 10% of the total training cases for model development.

For each iterative segmentation, the evaluation metrics include Dice Similarity Coefficient (DSC) and Normalized Surface Distance (NSD) to evaluate the segmentation region overlap and boundary distance, respectively. The final metrics used for the ranking are:

- DSC_AUC and NSD_AUC Scores: AUC (Area Under the Curve) for DSC and NSD is used to measure cumulative improvement with interactions. The AUC quantifies the cumulative performance improvement over the five click predictions, providing a holistic view of the segmentation refinement process. It is computed only over the click predictions without considering the initial bounding box prediction as it is optional.
- Final DSC and NSD Scores after all refinements, indicating the model's final segmentation performance.

In addition, the algorithm runtime will be limited to 90 seconds per class. Exceeding this limit will lead to all DSC and NSD metrics being set to 0 for that test case.

### 3.2   Implementation details

**Preprocessing** Following the practice in MedSAM [17], all images were processed to npz format with an intensity range of $[0, 255]$. Specifically, for CT

---

[1] A complete list is available at https://medsam-datasetlist.github.io/

images, we initially normalized the Hounsfield units using typical window width and level values: soft tissues (W:400, L:40), lung (W:1500, L:-160), brain (W:80, L:40), and bone (W:1800, L:400). Subsequently, the intensity values were rescaled to the range of $[0, 255]$. For other images, we clipped the intensity values to the range between the 0.5th and 99.5th percentiles before rescaling them to the range of $[0, 255]$. If the original intensity range is already in $[0, 255]$, no preprocessing was applied.

During training and inference, we first convert the grayscale images of the development set to RGB by duplicating the intensity values along the channel axis. We then resize the input images to a resolution of $512 \times 512$ without padding, and apply channel-wise RGB normalization to ensure consistency with the pretraining configuration of EfficientTAM.

**Environment settings** The development environments and requirements are presented in Table 1. Compared to the training on the 10% coreset, we use twice as many GPUs and the more powerful NVIDIA H100 for training on the full training dataset in order to accelerate the process.

**Table 1.** Development environments and requirements.

|  | coreset | all-data |
|---|---|---|
| System | Debian 12.11 LTS | |
| CPU | AMD EPYC 7543 32-Core @ 2.79 GHz | |
| RAM | 50G | 80G |
| GPU | $4 \times$ NVIDIA L40 48G | $8 \times$ NVIDIA H100 80G |
| CUDA version | 12.6 | |
| Programming language | Python 3.11 | |
| Deep learning framework | torch 2.6, torchvision 0.21 | |

**Training protocols** We fine-tune the pretrained EfficientTAM-S on the development set of the challenge, training all model components to maximize performance. As described in section 2.3, we selected a development set with a distribution of 3D imaging modalities similar to that of the validation set. For training on the full dataset, we oversample the underrepresented modalities MR, PET, and ultrasound by factors of 2, 3, and 3, respectively.

For each epoch, we randomly select one object per case. Starting from the middle slice, the model predicts with equal probability either from the middle toward the top or from the middle toward the bottom. If this segment spans more than 25 slices, a shorter subsection is randomly selected to prevent running out of memory. As described in section 2.2, we then simulate three correction clicks per object half in an iterative manner, resulting in a total batch size of four. We apply data augmentation to the training samples, including random horizontal flips and random affine transformations. The random affine transformations are

performed with a rotation angle in the range of $[-25, 25]$ and a shear angle in the range of $[-20, 20]$.

To supervise the model's predictions, we use a linear combination of focal loss and dice loss for the mask prediction, with a weighting ratio of 20:1. Compound loss functions have been shown to be robust in various medical image segmentation tasks [16]. We use the AdamW optimizer [15] with $\beta_1 = 0.9$, $\beta_2 = 0.999$, and a weight decay of 0.1. For the image encoder, we select a lower learning rate $(3.0 \times 10^{-6})$ compared to the other components $(5.0 \times 10^{-6})$, which require greater adaptation to the medical domain. Table 2 details the training protocols used for fine-tuning both the coreset and the all-data model.

Since evaluation on the validation set takes approximately 24 hours, we employed a binary search strategy to select the optimal model checkpoints. Assuming that performance improves with each epoch and only degrades in later stages due to overfitting, this approach allowed us to significantly reduce the number of required evaluation runs.

**Table 2.** Training protocols.

|  | coreset | all-data |
| --- | --- | --- |
| Pre-trained Model | EfficientTAM-S | |
| Data augmentation | Horizontal Flipping and Random Affine | |
| Batch size | 4 | |
| Input size | $512 \times 512 \times 3$ | |
| Optimizer | AdamW with weight decay set to 0.1 | |
| Initial learning rate (lr) | Img. enc.: 5e-6, other: 3.0e-6 | |
| Lr decay schedule | Cosine with an end value of 10% of the initial lr | |
| Loss function | $20 \times$ focal loss $+ 1 \times$ dice loss | |
| Number of model parameters | 34.1M | |
| Total epochs | 100 | 15 |
| Training time | 131 hours | 174 hours |
| Number of flops | 413.7P | 848.9P |

## 4 Results and discussion

### 4.1 Quantitative results on validation set

Table 3 compares the performance of the proposed model for the coreset track (iMedSTAM) with the baseline models on the public validation set. iMedSTAM consistently outperforms all baselines across most modalities and metrics. It achieves the highest average DSC_AUC (3.022), NSD_AUC (3.270), Final DSC (0.782), and Final NSD (0.847), indicating strong interactive segmentation performance and boundary accuracy. Even in the first iteration without correction clicks (DSC_1), it achieves the highest score of 0.684.

**Table 3.** Quantitative evaluation results of the validation set on the **coreset track**. DSC_AUC and NSD_AUC measure the cumulative improvement across interactions. Final DSC and NSD represent the scores after all refinements, while DSC_1 denotes the score obtained using only the initial prompt.

| Modality | Methods | DSC AUC | NSD AUC | DSC Final | NSD Final | DSC 1 |
|----------|---------|---------|---------|-----------|-----------|-------|
| CT | SAM-Med3D | 2.218 | 2.192 | 0.554 | 0.549 | 0.564 |
| | VISTA3D | 2.797 | 2.816 | 0.715 | 0.724 | 0.649 |
| | SegVol | 2.899 | 3.037 | 0.725 | 0.759 | **0.733** |
| | iMedSTAM | **3.227** | **3.327** | **0.828** | **0.856** | 0.730 |
| MRI | SAM-Med3D | 1.493 | 1.490 | 0.383 | 0.387 | 0.332 |
| | VISTA3D | 2.290 | 2.578 | 0.578 | 0.648 | 0.544 |
| | SegVol | 1.113 | 1.314 | 0.278 | 0.328 | 0.279 |
| | iMedSTAM | **2.771** | **3.209** | **0.724** | **0.837** | **0.629** |
| Microscopy | SAM-Med3D | 0.118 | 0.000 | 0.030 | 0.000 | 0.029 |
| | VISTA3D | 1.730 | **2.737** | 0.442 | **0.696** | **0.400** |
| | SegVol | 1.053 | 1.857 | 0.263 | 0.464 | 0.265 |
| | iMedSTAM | **1.731** | 2.197 | **0.474** | 0.604 | 0.334 |
| PET | SAM-Med3D | 2.104 | 1.789 | 0.528 | 0.449 | 0.525 |
| | VISTA3D | 2.388 | 2.098 | 0.612 | 0.543 | 0.561 |
| | SegVol | 2.968 | 2.856 | 0.742 | 0.714 | **0.739** |
| | iMedSTAM | **3.077** | **2.988** | **0.814** | **0.796** | 0.595 |
| Ultrasound | SAM-Med3D | 1.277 | 1.882 | 0.361 | 0.531 | 0.168 |
| | VISTA3D | 2.580 | 2.589 | 0.707 | 0.717 | 0.494 |
| | SegVol | 1.232 | 1.788 | 0.308 | 0.447 | 0.307 |
| | iMedSTAM | **3.623** | **3.595** | **0.912** | **0.908** | **0.888** |
| Average | SAM-Med3D | 1.813 | 1.822 | 0.460 | 0.466 | 0.430 |
| | VISTA3D | 2.529 | 2.667 | 0.645 | 0.682 | 0.587 |
| | SegVol | 1.963 | 2.147 | 0.491 | 0.537 | 0.494 |
| | iMedSTAM | **3.022** | **3.270** | **0.782** | **0.847** | **0.684** |

iMedSTAM ranks first in both DSC and NSD for CT, MRI, PET, and ultrasound. The model performs particularly well on ultrasound, achieving a Final DSC of 0.912, with a 21% margin over the second-best model, significantly outperforming all other methods. Only for the challenging microscopy images does VISTA3D surpass iMedSTAM in NSD_AUC and Final NSD. It is also noteworthy that while SegVol achieves impressive DSC values for CT and PET with just a single prompt, it shows little improvement with additional correction clicks.

Table 4 shows the performance on the validation set of the all-data track, where our model, iMedSTAM, demonstrates strong and consistent results. By exposing the models to the full development set, all models achieve a performance improvement. iMedSTAM achieves the best scores across all metrics for the MRI and ultrasound modalities. In contrast, nnInteractive performs better on CT and PET 3D medical images. Similar to the coreset track, iMedSTAM shows weaknesses in segmenting microscopy images. In this category, VISTA3D achieves high NSD scores, while nnInteractive yields better DSC values. Overall, iMedSTAM attains the highest average NSD_AUC (3.323) and DSC_1 (0.702)

across all modalities. For DSC_AUC, Final DSC, and Final NSD, nnInteractive leads slightly, with margins of 0.002, 0.12, and 0.007, respectively. These results indicate that nnInteractive is more effective in incorporating refinement clicks.

**Table 4.** Quantitative evaluation results of the validation set on the **all-data track**. DSC_AUC and NSD_AUC measure the cumulative improvement across interactions. Final DSC and NSD represent the scores after all refinements, while DSC_1 denotes the score obtained using only the initial prompt.

| Modality | Methods | DSC AUC | NSD AUC | DSC Final | NSD Final | DSC 1 |
|---|---|---|---|---|---|---|
| | VISTA3D | 3.169 | 3.265 | 0.804 | 0.834 | 0.751 |
| CT | nnInteractive | **3.434** | **3.574** | **0.876** | **0.916** | **0.767** |
| | iMedSTAM | 3.277 | 3.389 | 0.838 | 0.869 | 0.751 |
| | VISTA3D | 2.589 | 2.968 | 0.654 | 0.749 | 0.619 |
| MRI | nnInteractive | 2.698 | 3.029 | 0.730 | 0.823 | 0.541 |
| | iMedSTAM | **2.824** | **3.265** | **0.735** | **0.848** | **0.646** |
| | VISTA3D | 2.120 | **3.226** | 0.548 | **0.824** | **0.486** |
| Microscopy | nnInteractive | **2.331** | 3.111 | **0.594** | 0.789 | 0.480 |
| | iMedSTAM | 1.690 | 2.347 | 0.467 | 0.658 | 0.354 |
| | VISTA3D | 2.640 | 2.400 | 0.678 | 0.623 | 0.608 |
| PET | nnInteractive | **3.348** | **3.324** | **0.855** | **0.849** | **0.719** |
| | iMedSTAM | 3.001 | 2.895 | 0.789 | 0.766 | 0.592 |
| | VISTA3D | 2.866 | 2.844 | 0.810 | 0.808 | 0.509 |
| Ultrasound | nnInteractive | 3.348 | 3.324 | 0.855 | 0.849 | 0.580 |
| | iMedSTAM | **3.641** | **3.633** | **0.915** | **0.916** | **0.896** |
| | VISTA3D | 2.858 | 3.074 | 0.729 | 0.786 | 0.670 |
| Average | nnInteractive | **3.069** | 3.285 | **0.803** | **0.864** | 0.647 |
| | iMedSTAM | 3.067 | **3.323** | 0.791 | 0.857 | **0.702** |

To better understand how the AUC values are derived, Fig. 3 illustrates the performance progression of all models with respect to the number of correction clicks. For all models, segmentation accuracy improves incrementally with each additional correction click. The only exception is SegVol, which processes only bounding boxes as input and therefore cannot benefit from correction clicks. Both the all-data and coreset versions of iMedSTAM achieve the highest average DSC and NSD with only a single initial user prompt and no correction clicks. It is only after the second refinement click for DSC and the third for NSD that nnInteractive surpasses iMedSTAM.

Overall, we observe that the initial correction clicks contribute the most to performance gains, while subsequent clicks yield diminishing improvements. Furthermore, the coreset version of iMedSTAM initially performs slightly worse than the all-data version but gradually closes the gap with additional clicks, eventually achieving nearly comparable performance. This suggests that fine-tuning with more data enables faster initial object recognition. However, the coreset is sufficient for the model to learn how to handle and interpret correction clicks.

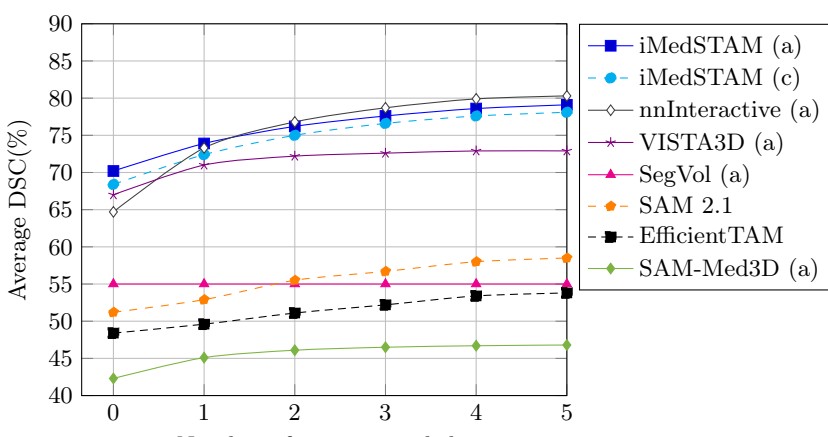

**Fig. 3.** Interactive segmentation performance on the validation set with respect to the number of correction clicks. Models trained on the full development set are marked with (a), while those trained only on the coreset are marked with (c).

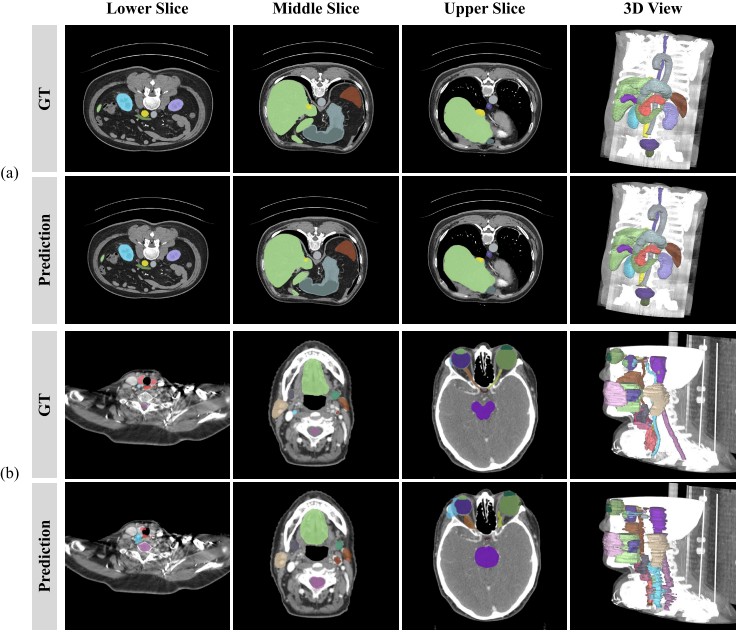

**Fig. 4.** Qualitative results for CT on the validation set. We illustrate one well-segmented case (a) and one challenging case (b). The model achieves a DSC of 90% in case (a) and 56 % in case (b).

## 4.2  Qualitative results on validation set

Fig. 4 and  5 illustrate CT and MRI examples where iMedSTAM performs well, along with one example for each modality where its performance is poor. For each case, we visualize the ground truth and the model's predicted masks.

In the CT example (a), iMedSTAM accurately segments and differentiates various abdominal organs throughout the entire volume. As seen in the middle slice, the model handles the segmentation of the liver (light green), even when it consists of several separate regions in that plane. It also successfully identifies the inferior vena cava (yellow), which is partially encircled by the liver. Overall, a DSC of 90% is achieved across all segmented objects. In contrast, iMedSTAM struggles in the CT case (b). The 3D view clearly shows that only the lower portion of the brainstem (cyan) is detected, while the structure is lost toward the middle and upper slices. A likely reason is the very narrow diameter of the brainstem in the middle, which may be misinterpreted as the end of the object. Additionally, many of the predicted objects exhibit non-smooth surfaces, with visible edges arising from the model's slice-by-slice processing. A further negative factor is the greater physical distance between slices compared to case (a). As a result, the objects exhibit more pronounced discontinuities in their location from one slice to the next.

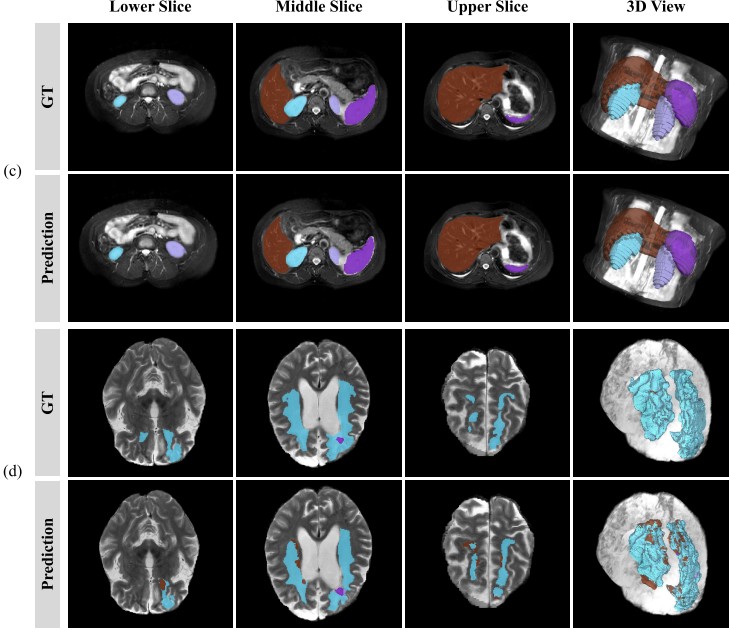

**Fig. 5.** Qualitative results for MRI on the validation set. We illustrate one well-segmented case (c) and one challenging case (d). The model achieves a DSC of 96% in case (c) and 37% in case (d).

iMedSTAM is able to segment all four visible abdominal organs in the MRI of case (c) with high accuracy, achieving an above-average DSC of 96%. In contrast, the model reaches only a DSC of 37% in case (d). While the surrounding non-enhancing FLAIR hyperintensity (cyan) is segmented reasonably well despite its complex structure, the mask for the non-enhancing tumor core (brown) is significantly oversegmented. One contributing factor is that, in this case, only a single point prompt was provided instead of an initial bounding box. This means the model lacked clear spatial boundaries across slices and had to infer them on its own. Additionally, the original resolution of the image is quite low at 218 × 182 pixels. When resized, the image becomes slightly blurry, further obscuring the color boundaries between different structures.

Fig. 6 shows one PET example where the model performs well and another where it encounters difficulties. Compared to CT and MRI, PET images use a different color scale and tend to appear more point-based than continuous. In case (e), this does not pose a problem, as the various lesions are relatively small. In contrast, in case (f), iMedSTAM significantly oversegments the brown lesion in the lower slice, leading to a reduced overall performance. Even though the other lesions are segmented accurately, the model achieves only a DSC of 53% in this case.

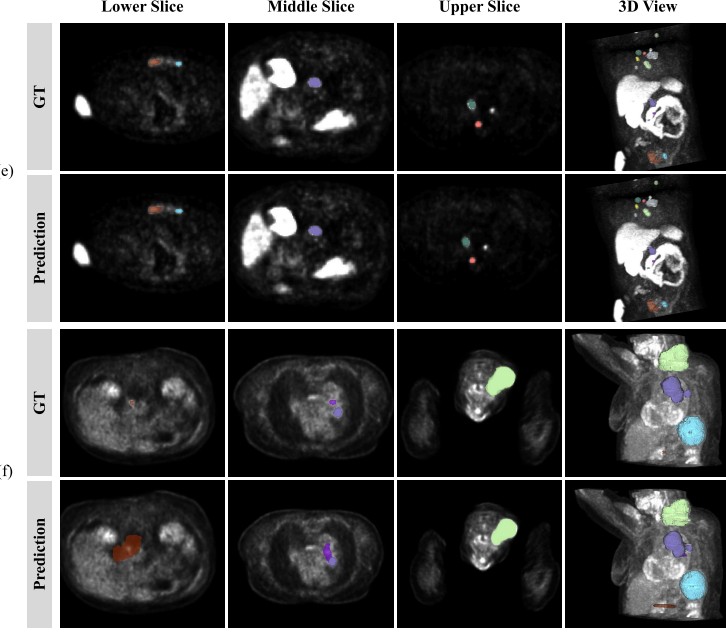

**Fig. 6.** Qualitative results for PET on the validation set. We illustrate one well-segmented case (e) and one challenging case (f). The model achieves a DSC of 89% in case (e) and 53% in case (f).

Examples of the microscopy and ultrasound modalities are shown in Fig. 7. In the microscopy case (g), iMedSTAM successfully tracks a total of 90 distinct Alzheimer's disease plaques in the brain across 122 slices, achieving a DSC of 72%. In case (h), brain vessels are observed under the microscope. In the ground truth, two large vascular networks are each grouped into a single class. Since these classes consist of many disjoint areas, it is challenging for the model to establish coherent segmentations. Additionally, similar to the MRI case (d), no bounding box was provided, and only a single positive point click was used as the initial prompt, making the segmentation task even more difficult.

In case (i), ultrasound recordings of the heart are shown. The model successfully tracks and segments the left ventricle, myocardium, and left atrium almost perfectly throughout the entire sequence. In case (j), a lower leg is depicted. The model struggles only with the musculus gastrocnemius (purple) in the initial frames, where it fails to shrink the segmentation quickly enough before the structure moves out of the frame. As already observed in the quantitative evaluation, iMedSTAM performs very well on the ultrasound modality, achieving a DSC of 83% even in the worst case.

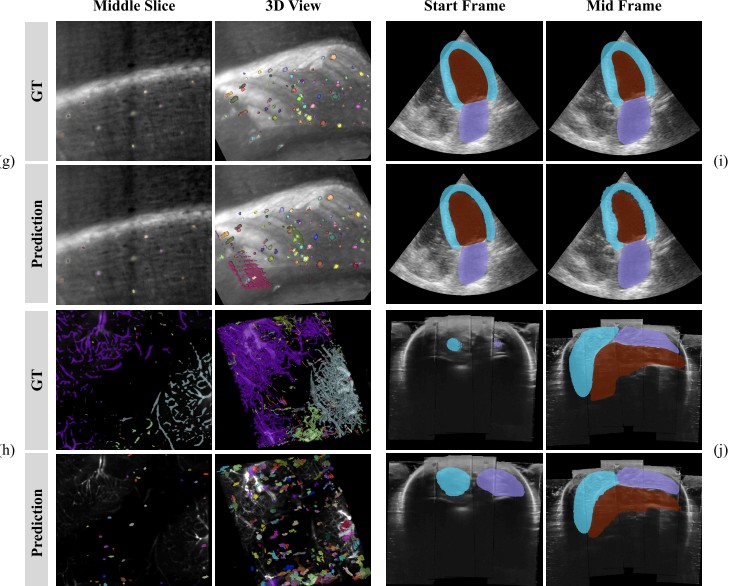

**Fig. 7.** Qualitative results for microscopy and ultrasound on the validation set. For each modality, we illustrate one well-segmented case (g/i) and one challenging case (h/j). The model achieves a DSC of 72% in case (g), 22% in case (h), 95% in case (i), and 83% in case (j).

**Table 5.** Results and ranking of the top five teams on the final testing set.

| Team | Rank | DSC AUC | NSD AUC | DSC Final | NSD Final | Track |
|------|------|---------|---------|-----------|-----------|-------|
| aim (ours) | 1 | 3.129 | 3.258 | 0.805 | 0.842 | all-data |
| yiooo [36] | 2 | 2.981 | 3.009 | 0.767 | 0.779 | all-data |
| norateam [20] | 3 | 2.911 | 2.970 | 0.754 | 0.775 | coreset |
| sjtu-426lab [8] | 4 | 2.552 | 2.645 | 0.640 | 0.664 | all-data |
| lexor [2] | 5 | 2.501 | 2.572 | 0.625 | 0.643 | coreset |

### 4.3   Results on final testing set

On the final testing set, our team aim achieved the highest ranking in both the all-data and coreset tracks. As shown in Table 5, iMedSTAM attained final DSC and NSC scores of 0.805 and 0.842, respectively. For DSC_AUC and NSD_AUC, we obtained scores of 3.129 and 3.258, demonstrating significant superiority across all four evaluation metrics. The second-best performing team, yiooo, proposed a dual-expert architecture integrating both global and local Region-of-Interest strategies [36], while the third-ranked team employed a fine-tuned nnInteractive model [20].

### 4.4   Limitation and future work

iMedSTAM has demonstrated significant improvements, particularly when compared to the baseline models in the coreset track. However, certain limitations remain. One notable challenge lies in the microscopy modality. Since the validation set includes generally difficult cases, none of the evaluated models achieve outstanding results in this modality. Nevertheless, iMedSTAM lags slightly behind VISTA3D and nnInteractive. This performance gap, compared to the other modalities, may be attributed to the fact that microscopy is substantially underrepresented in the development set, accounting for only 0.39%. Future work could address this by collecting additional microscopy cases and incorporating them into the training process.

Another limitation is iMedSTAM's performance on 3D medical images with relatively large physical spacing between slices. In such cases, the model may encounter more abrupt changes in object position from slice to slice than typically occur in video data, which the model was originally pre-trained on. As a result, this can lead to non-smooth surface segmentations. True 3D native models generally do not face this issue, as they are better at capturing spatial continuity in these scenarios. A potential approach to mitigate anisotropy in 3D medical images is to integrate an ASRGAN for inter-slice recovery [5] or to apply slice imputation with a synthesis model [33]. While such methods can restore spatial continuity, they substantially increase execution time due to the processing of artificially generated slices.

Furthermore, a modality-aware approach could be explored, where separate models are fine-tuned for each imaging modality rather than using a single

general-purpose model. In real-world applications, users often know the modality of the input data in advance, and leveraging this information could enhance segmentation performance.

Expanding the training data to include additional publicly available datasets such as CAMUS [14], EMIDEC [13], SpineMets [21], and AortaSeg24 [9], which were not included due to redistribution and licensing constraints, could further improve generalization and robustness across modalities. Moreover, optimizing the inference pipeline by implementing intelligent caching of image embeddings across interaction iterations and object-level passes could reduce computational redundancy and further improve efficiency.

## 5   Conclusion

In this work, we presented iMedSTAM, based on the EfficientTAM architecture. iMedSTAM is capable of interactively segmenting not only medical videos but also delivering accurate segmentations across multiple slices in 3D imaging modalities such as CT, MRI, and PET. On the coreset track, our model significantly outperforms baseline models across all modalities, and on the all-data track, it performs nearly on par with the recent nnInteractive model. Furthermore, through iterative correction clicks, users can progressively refine segmentation quality using iMedSTAM. Our best-performing model achieves an average final DSC and NSD of 0.791 and 0.857, respectively. This performance suggests that it could also be utilized in the construction of additional 3D medical image and video datasets, similar to the approaches in [23] and [19], in order to significantly accelerate the annotation process.

**Acknowledgements**  We thank all the data owners for making the medical images publicly available and CodaLab [35] for hosting the challenge platform. This research project is funded by the Bavarian Research Institute for Digital Transformation (bidt), an institute of the Bavarian Academy of Sciences and Humanities. Additionally, this work was supported by the bayerischen tschechischen Hochschulagentur (BTHA) under grant BTHA-JC-2024-52. The author is responsible for the content of this publication.

**Code availability**  The training and inference scripts, along with the trained models, have been made publicly available at https://github.com/tofriede/iMedSTAM.

**Disclosure of Interests**  The authors have no competing interests to declare that are relevant to the content of this article.

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

# Appendix

**Table 6.** Quantitative evaluation results of different SAM2 and EfficientTAM versions on the validation set. All models were not fine-tuned on the development set and thus reflect their default performance. We demonstrate how the number of parameters and input resolution affect the average runtime per object and segmentation quality.

| Model | Input Resolution | Parameters | Running Time (s) | DSC AUC | NSD AUC | DSC Final | NSD Final |
|---|---|---|---|---|---|---|---|
| SAM 2.1 Tiny | 1024 × 1024 | 38.9M | 5.017 | 2.281 | 2.364 | 0.594 | 0.617 |
| SAM 2.1 B+ | 1024 × 1024 | 80.8M | 5.673 | 2.251 | 2.348 | 0.585 | 0.612 |
| EfficientTAM S | 1024 × 1024 | 34.1M | 4.705 | 2.157 | 2.259 | 0.556 | 0.584 |
| EfficientTAM S | 512 × 512 | 34.1M | 2.248 | 2.082 | 2.162 | 0.538 | 0.555 |

**Table 7.** Incremental performance improvement by adding each component step by step, evaluated on the public validation set. The baseline is an EfficientTAM-S model with an input resolution of 512 × 512, which receives the initial prompt in the first slice/frame where the object appears and propagates the mask in one direction only. All subsequent variants receive the initial prompt in the middle slice and propagate the mask bidirectionally. Fine-tuning was performed on the full development set.

| Model Variant | DSC AUC | NSD AUC | DSC Final | NSD Final |
|---|---|---|---|---|
| EfficientTAM S (512 × 512) | 1.687 | 1.702 | 0.324 | 0.299 |
| + Bidirectional mask propagation | 2.082 | 2.162 | 0.538 | 0.555 |
| + Conditioning on all correction slices/frames | 2.175 | 2.413 | 0.577 | 0.642 |
| + Restricting conditioning to the same direction | 2.241 | 2.432 | 0.596 | 0.648 |
| + Non-overlapping post-processing step | 2.423 | 2.550 | 0.634 | 0.673 |
| + Fine-tuning on the development set | 3.067 | 3.323 | 0.791 | 0.857 |