# OpenReview forum: "iMedSTAM: Interactive Segmentation and Tracking Anything in 3D Medical Images and Videos"
_thecvf.com/CVPR/2025/Workshop/MedSegFM — CVPR 2025 Workshop MedSegFM Submission_

### Official Review · Reviewer_dcxQ · 2025-09-08
**Promising Slice-Based 3D Segmentation with Competitive Results but Limited Methodological Detail**

**Rating:** 8
**Confidence:** 4

**Review:**

# Review

The authors propose an interactive 3D segmentation model building on EfficientTAM, processing 3D images on a per-slice basis and memory cross-attention. The model is evaluated on the hidden test set, obtaining very competitive results.

## Strengths
- A relatively thorough evaluation, with results per iterative refinement iteration presented
- Besides the EfficientTAM model, the authors also evaluate SAM2.1, indicating it might be better performing but at a higher computational cost.
- Promising results suggest that per-slice processing might be a feasible approach for clinical use.

## Weaknesses
- Limited description and discussion of incorporation of user prompts, and implementation of cross-attention implementation.
- Lacks discussion of clinical applicability, how far away are we from clinical usefulness?

## Novelty & Significance
The paper's idea is moderately novel within this particular use case but incorporates architectures that have been explored in related domains.

## Clarity
The work is presented in an understandable manner, but some more specificity could be needed in certain parts of the methodology section.

## Suggestions for Improvement and Detailed Comments
- Please help the reader by writing full model names or authors and not only citations. (examples: "as shown in [23], [35]", "we adopt a video-based approach similar to [16] and [24]")
- When describing the model architecture, it would be helpful if some details on the model architecture were provided. Examples include: embedding dimension, shapes of encoded images, position encoding method, how are boxes encoded (2 embeddings? 4?).
- You go from single-channel to 3-channel images to ensure compatibility with the pretrained image model - how is this done? Please discuss whether this is optimal or not.
- You mention Microscopy being a 2D+t modality, I am not sure this is correct (in this development set).
- When showing example segmentations, consider zooming in to the region of interest, as it is difficult to see for example some of the CT results.
- You mention large physical spacing as a problem for your model (limitations). On the contrary, per-slice based models might have good anisotropy or varying frame rates via the attention mechanism. Perhaps it is worth discussing how this could be done.

---

> ### Author Response · Authors · 2025-10-06
>
> We thank the reviewer for the constructive and detailed feedback, as well as for highlighting the strengths of our work. We have carefully considered the suggestions and revised the manuscript accordingly:
>
> - We now provide full model names or author names in all in-text citations.
> - In Section 2.2, we added more details on the model architecture, including the image encoder, image embedding dimensions, and the encoding of box prompts.
> - In Section 3.2 (Preprocessing), we clarified how the grayscale images from the development set are converted into RGB images.
> - We corrected passages that previously suggested microscopy is a 2D+t modality. Furthermore, we improved Figure 7 by including a 3D view for the microscopy examples.
> - In Figure 4 (CT examples), we zoomed in on the regions of interest to make the segmentations more clearly visible.
> - In Section 4.4 (Limitations), we extended the discussion with a new paragraph explaining that the segmentation issues caused by anisotropy in 3D medical images could be reduced by applying inter-slice imputation.

---

### Official Review · Reviewer_BeQQ · 2025-10-12
**iMedSTAM: Strong, Practical 3D Interactive Segmentation---Great Results, More Analysis Would Seal the Deal**

**Rating:** 7
**Confidence:** 4

**Review:**

$\textbf{Overall Review:}$

This paper proposes $\textrm{iMedSTAM}$, an extension of a video-style interactive segmentation framework to 3D medical images and videos. It supports box/point prompts and iterative corrections, uses a lightweight memory module with bi-directional slice propagation, and adds a non-overlap post-processing step based on logits to handle multiple objects. Across public benchmarks and multiple modalities, the system beats or matches strong baselines, and it ranked first on the final test sets under both tracks. The engineering is clean and reproducible, with clear practical value. That said, the paper would benefit from deeper ablations and clearer reasoning around several core design choices (memory length, bi-directional policy, prompt strategy). The method’s “2.5D” nature also leaves some open questions about boundary smoothness and robustness for thick-slice cases. Overall: solid, well-executed work with room to push on analysis and novelty claims.

$\textbf{Strengths:}$

$\textrm{1. Right problem, practical interface.}$ Bringing interactive video segmentation ideas to 3D clinical data is timely. The system supports common prompt types (points/boxes) and multi-round edits, which maps nicely to real annotation and correction workflows.

$\textrm{2. Simple, effective propagation + memory.}$ Starting from a middle slice and propagating in both directions, while only using same-direction memory, is a neat way to avoid ``memory pollution.'' A small FIFO memory (e.g., last few slices) gives a good speed/quality tradeoff.

$\textrm{3. Useful post-processing.}$ The non-overlap mechanism works on mask logits rather than relying on processing order, and caching helps cut redundant compute across interaction rounds.

$\textrm{4. Robust across modalities; ultrasound stands out.}$ Results are consistently strong, with especially high scores on ultrasound and competitive numbers on MRI.

$\textrm{5. Systematic experiments and clear visuals.}$ The paper reports final metrics, AUC vs.\ click count, and diverse qualitative examples, making it easy to follow the evidence.

$\textrm{6. Efficiency and practicality.}$ With an EfficientTAM backbone and $512{\times}512$ inputs, latency is noticeably lower than heavier baselines, even without fine-tuning. Engineering details (caching, prompt handling) are clearly described.

$\textrm{7. Reproducibility.}$ Data processing, losses/optimizers, checkpoint selection, and code pointers are documented well, making it feasible to re-run and extend.


$\textbf{Weaknesses:}$

$\textrm{1. Novelty feels incremental.}$ The core idea is a careful, well-engineered adaptation of existing interactive video segmentation to 3D/medical. That’s valuable, but the paper should sharpen what’s fundamentally new vs.\ prior SAM/SAM2/MedSAM2/interactive variants.

$\textrm{2. Inherent ``2.5D'' limits.}$ The paper itself notes less-smooth boundaries and slice-to-slice discontinuities, especially with thick slices. A head-to-head against true 3D encoders (or 3D adapters) would help quantify this gap.

$\textrm{3. Missing key ablations.}$ There’s no tight breakdown for memory window size, starting-in-the-middle vs.\ other policies, same-direction memory vs.\ all-memory, the non-overlap step, or input resolution (e.g., $512$ vs.\ $1024$). It’s hard to attribute which choices matter most.

$\textrm{4. Prompt strategy underexplored.}$ Beyond the global AUC trends, we don’t see a deeper look at box vs.\ points, balance of positive/negative clicks, or click-order effects per anatomy.

$\textrm{5. Microscopy underperformance not dissected.}$ The authors mention tiny training share for that modality, but don’t test re-sampling, style transfer, or modality-specific fine-tuning to validate the hypothesis.

$\textrm{6. Potential eval biases.}$ Choosing checkpoints via validation can overfit that split; end-to-end latency under multi-object, multi-round interactions (with the 90s/class constraint) could use fuller statistics.

$\textrm{7. Some engineering choices lack sensitivity checks.}$ CT window settings, always resizing to $512{\times}512$ (and the geometric distortion it implies), and the effect of downsampling/zero-padding aren’t quantified.


$\textbf{Suggestions:}$

$\textrm{1. Do the full ablation/attribution pass.}$ Quantify contributions of (i) memory window length; (ii) middle-start \& bi- vs.\ uni-directional propagation; (iii) same-direction-only memory; (iv) removing or swapping the non-overlap step; (v) input resolution vs.\ latency/memory. Report impacts on DSC/NSD and AUC.

$\textrm{2. Analyze prompt cost vs.\ gain.}$ Compare box vs.\ points, pos/neg ratios, and click order. Given the 90s/class limit, show a ``gain-per-second'' curve to guide real-world usage.

$\textrm{3. Benchmark against true 3D options.}$ Add fair baselines with 3D encoders or lightweight 3D adapters, focusing on thick-slice and highly deformable cases. Measure boundary smoothness/NSD continuity explicitly.

$\textrm{4. Modality-aware training.}$ Try modality-specific heads or fine-tuning for microscopy/PET, plus data balancing, style transfer, or synthetic augmentation. Report cross-modality transfer costs/benefits.

$\textrm{5. Post-processing and calibration.}$ Quantify the non-overlap step’s lift and consider temperature calibration or logit normalization so multi-class confidence is comparable.

$\textrm{6. Deeper error analysis.}$ Break down failure modes by anatomy complexity, inter-slice motion, contrast, resolution, and prompt type, then propose targeted fixes (e.g., inter-slice smoothness priors, shape/surface consistency).

$\textrm{7. Operational metrics for deployment.}$ In realistic multi-object, multi-round sessions, report mean/percentile latency and peak memory. Include scalability curves (GPUs, resolutions, caching on/off) for easy adoption.

$\textrm{8. Stronger checkpoint policy.}$ Replace validation-driven search with fixed milestones, a separate dev split, or cross-validation to reduce overfitting risk.

---

> ### Author Rebuttal · Authors · 2025-11-03
>
> We thank the reviewer for the valuable and constructive feedback, which helped us improve the clarity, completeness, and presentation of our manuscript. We have carefully considered the suggestions and revised the manuscript accordingly:
>
> - We have added the results of our ablation study to the Appendix (Table 7), which details how each component of iMedSTAM incrementally contributes to the final performance. The table shows the quantitative impact on all four evaluation metrics.
> - Although we do not present a direct "gain-per-second curve", as the runtime strongly depends on the object size along the third dimension, Figure 3 illustrates how performance improves with the number of corrective clicks. Combined with the average runtime of the EfficientTAM-S model across all iterations (Table 6), this provides a clear indication of efficiency per additional corrective click, as the per-iteration runtime remains nearly constant.
> - Training modality-specific heads was explicitly prohibited in the challenge setting, as the focus was on developing a foundation model. However, we have added a statement in the Limitations section (4.4) noting that modality-specific fine-tuning could potentially improve performance in real-world applications and represents an interesting direction for future work.
> - The performance gain resulting from the non-overlapping post-processing step is now included in Table 7 of the Appendix.
> - As also suggested by Reviewer 1, we have added a dedicated discussion in Section 4.4 (Limitations) describing how segmentation issues caused by anisotropy in 3D medical images could potentially be addressed in future work through improved spatial consistency mechanisms.
> - The average runtime performance for the segmentation of a single object, including all iterative refinement rounds, is reported in Table 6. The results were obtained using an NVIDIA L40 GPU with the EfficientTAM S (512 × 512) configuration.
> - For model selection, we relied on performance on an unseen validation set, which is a widely accepted practice in machine learning research. Using a fixed milestone such as the last epoch could increase the risk of overfitting to the training data. Cross-validation was not feasible due to the dataset size and the training time exceeding one week per run. Nevertheless, the model was evaluated on an independent test set, where it achieved comparable performance, indicating a low risk of overfitting to the validation set.

---

### Decision · Program_Chairs · 2025-11-12

Accept